# Optimum Capacity and Placement of Storage Batteries Considering Photovoltaics

**Hiroki Aoyagi** [1,*], **Ryota Isomura** [1,†], **Paras Mandal** [2,†] , **Narayanan Krishna** [3,†] , **Tomonobu Senjyu** [1,†] , **and Hiroshi Takahashi** [4,†]

1   Faculty of Engineering, University of the Ryukyus, 1 Senbaru, Nishihara-cho, Nakagami, Okinawa 903-0213, Japan; e155508@eve.u-ryukyu.ac.jp (R.I.); b985542@tec.u-ryukyu.ac.jp (T.S.)
2   Department of Electrical and Computer Engineering, Power and Renewable Energy Systems (PRES) Lab, University of Texas at El Paso, El Paso, TX 79968, USA; pmandal@utep.edu
3   Department of Electrical and Electronics Engineering, SASTRA Deemed University, Thanjavur 613401, India; narayanan@eee.sastra.edu
4   Fuji Elctric Co., Ltd., Tokyo 141-0032, Japan; takahashi-hirosi@fujielectric.com
*   Correspondence: k188540@eve.u-ryukyu.ac.jp; Tel./Fax: +81-98-895-8686 (ext. 8686)
†   These authors contributed equally to this work.

**Abstract:** In recent years, due to the enforcement of the Feed-in tariff (FIT) scheme for renewable energy, a large number of photovoltaic (PV) has been introduced, which causes fluctuations in the supply-demand balance of a power system. As measures against this, the introduction of large capacity storage batteries and demand response has been carried out, and the balance between supply and demand has been adjusted. However, since the increase in capacity of the storage battery is expensive, it is necessary to optimize the capacity of the storage battery from an economic point of view. Therefore, in the power system to which a large amount of photovoltaic power generation has been introduced, the optimal capacity and optimal arrangement of storage batteries are examined. In this paper, the determination of storage battery placement and capacity considering one year is performed by three-step simulation based on probability density function. Simulations show the effectiveness of storage batteries by considering the introduction of demand response and comparing with multiple cases.

**Keywords:** demand response; photovoltaic power systems; storage battery; unit commitment

## 1. Introduction

Recently, the introduction of renewable energy sources (RES) has been increasing significantly due to the enforcement of Feed-in tariff (FIT) and the environmental impact of human activities. However, the output of these types of power sources is greatly influenced by natural phenomena such as solar radiation and wind speed, making it difficult to predict their power generation. In Japan, RES, especially photovoltaic (PV) power generation, has been introduced on a large scale which further increases the influences due to the large generated power error which causes the fluctuations of system voltage and supply-demand balance.

In recent years, there are many researchers dealing with such renewable energy problems [1–6]. Furthermore, conventional day-ahead unit commitment (UC) is insufficient for safe operation of power systems. Therefore, it is necessary to consider the uncertainty of output fluctuation when a large amount of renewable energy electric power system is introduced. Various optimization problems have been solved to cope with uncertainty due to renewable energy [7–12]. Moreover, when predicting renewable energy output power on the previous day and making an operation

plan, the prediction error will increase with the passage of time. The operation plan made on the previous day is not possible to deal with forecasting error. In this way, with the massive introduction of renewable energy power generation, it becomes difficult to predict power generation of the power grid, therefore the consideration of uncertainty becomes an important issue. Moreover, it is necessary to develop a model to balance demand supply including prediction error. Also, it will be necessary to develop an effective approach to deal with the power generation prediction error of renewable energy generation facilities. Conventionally, the power fluctuations caused by renewable energy generation facilities have been compensated by thermal generators. However, there are restrictions on the adjustable amount (up/down spinning reserves). When power fluctuations exceeding the limit of the adjustable amount flow into the power system, the fluctuations of system frequency and system voltage are caused. As a countermeasure for that, the introduction of large capacity storage batteries and load demand adjustment (demand response) are considered [13–15]. It is possible to adjust the disturbance of demand-supply balance due to prediction error of generated power from renewable energy generation facilities by introducing storage batteries and demand response. Furthermore, it is also possible to reduce operating costs by cooperative control of thermal power generator, storage battery and demand response. However, since increasing the storage capacity of the storage battery is expensive, it is necessary to optimize the capacity of the storage battery from the economical point of view. The determination of optimum placement and the optimum capacity of the storage battery have been solved by various methods. Y. Zheng et al. have used a cost-benefit analysis method aimed at maximizing distribution companies (DISCO's) profit from energy transactions, system planning, and operational cost savings [16]. Mostafa Nick et al. solve the problem of minimizing the storage battery installation-cost by using ADMM (Alternating Direction Method of Multipliers) [17]. C. H. Lo et al. have solved the optimization problem by an algorithm combining multi-path dynamic programming (MPDP) and time-shift technique [18]. S. Kahrobaee et al. proposed a hybrid stochastic method based on Monte Carlo simulation and particle swarm optimization, and decided the optimum size of wind power generation and storage battery [19]. References [20,21] have optimized capacity of storage batteries using bilevel program. References [22,23] introduce various situations, optimum operation of storage batteries in various types of storage batteries, and methods of determining the size of the storage battery. In addition, these papers have sized the storage battery by linear problem [24,25].

In this paper, we solved Mixed-integer linear programming (MILP) problem dealing with DC power flow calculation. Moreover, when determining the demand response and the capacity of the storage battery, it is necessary to consider the adjustable amount in the case where the generated power of the renewable energy generation facility greatly fluctuates. In this paper, the prediction error is quantified using the machine learning model which is the most accurate, and the spinning reserve that can correspond to the forecasting error which is decided on the previous day. Also, in the power grid where solar power generation equipment was massively introduced, we study the optimum capacity and optimum arrangement of grid storage batteries based on the dynamic operation method. This study reports, an economically useful planning method that does not involve rapid and significant operation change of thermal power generations via dynamically planning every 3 h. The effectiveness of the proposed method is verified through one-year simulation of transmission systems using MATLAB software.

In this paper, we propose a new method of determining the optimal arrangement and capacity for storage batteries. By dividing the simulation into three stages, we determined the optimal placement and capacity of the storage battery. This is derived from the idea of the three-stage battery capacity determination method in the literature [26]. The first stage predicts PV output. In the second stage, taking into account the PV prediction error, the optimum capacity and the optimum number of storage batteries are obtained from the standard deviation. In the third stage, a one-year simulation is conducted to confirm the effectiveness of the determined battery placement and capacity. By optimizing storage battery capacity and layout, it is possible to calculate the optimum storage battery capacity for economic improvement and introduce the optimum storage battery capacity, so that even when a large

amount of renewable energy equipment is introduced, economic efficiency indicates that operations that do not deteriorate can be performed. Also, we show that proposed method can achieve operation which minimizes fuel cost and startup cost of the generator, and minimizes operation cost.

The paper is organized as follows: Section 2 explains the methodology for determining the optimal placement and optimal capacity of the storage battery proposed in this paper, the objective function for solving the simulation, and the formulation of constraints. Section 3 explains the conditions for simulation to confirm the effectiveness of the optimal placement and capacity of the storage battery determined in Section 2. In Section 4, the simulation results are presented as case studies. Section 5 concludes this paper.

## 2. Problem Formulations

Figure 1 shows the method for determining the capacity and location of the storage battery [26]. In the first stage, PV output forecasting is performed based on the data of 2015 using feedforward neural network (FFNN) which is machine learning. In the second stage, the UC problem will be solved on a daily basis in the 2016 load demand, using the optimal placement and capacity of storage batteries as variables. In addition, the decision variable $SoC_b^{max}$ is calculated for one year. It is assumed here that the power demand forecast for 2016 is accurate. In addition, let standard deviation $\sigma_b^{SOC}$ of $SoC_b^{max}$ for 1 year be the storage battery optimal capacity in the bus $b$. In the third stage, a one-year simulation is conducted using the load demand of 2017 in order to confirm the effectiveness of the determined placement and capacity of storage batteries. The decision variable $SoC_b^{max}$ is calculated for one year from the one-day optimization problem with the following objective function and constraints. Let the standard deviation $\sigma_b^{SOC}$ of $SoC_b^{max}$ for one year be the optimum capacity of the bus $b$.

$$\sigma_b^{SOC} = \sqrt{\frac{1}{n}\sum_{i=1}^{n}(SoC_{b,i}^{max} - \overline{SoC_{b,i}^{max}})^2} \tag{1}$$

$$\overline{SoC_{b,i}^{max}} = \frac{\sum_{i=1}^{n} SoC_{b,i}^{max}}{n} \tag{2}$$

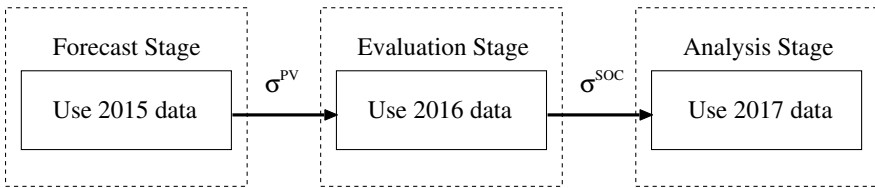

**Figure 1.** Optimal placement and optimum capacity.

### 2.1. Objective Function

The objective function Equation (3) is minimizing total cost ($F$) which includes fuel cost ($FC$), start-up cost ($SUC$) and penalty cost of energy not served ($ENS \times C_{ens}$).

$$F = \sum_{i=1}^{N_G}\sum_{t=1}^{24}\sum_{b=1}^{N_B}(FC_{itb} + SUC_{itb}) + \sum_{t=1}^{24}(ENS_t \times C_{ens}) + \sum_{b=1}^{N_B}(SoC_b^{max} \times IC_{SoC}) \tag{3}$$

$IC_{SoC}$ is the installation cost of storage batteries converted in one day.

$$IC_{SoC} = \frac{C_{SoC}}{h \times 365} \tag{4}$$

## 2.2. Constraints

- System power balance
  Equation (5) is a constraint that the sum of the power demand $D_{tb}^{net}$ and the amount of electricity generated by the generator $P_{itb}$ and $ENS_t$ become equal.

$$\sum_{i=1}^{N_G}(P_{itb} \times X_{itb}) + ENS_{tb} = D_{tb}^{net} \quad \forall t \in T, \; \forall b \in B \tag{5}$$

- Generation output limits
  The generation limits of each unit for each period are set as follows [27]

$$P_i^{min} \leq P_{it} \leq P_i^{max} \tag{6}$$

  Constraints Equation (6) bound the generation by the minimum power output and the maximum available power output of unit $i$ in period $t$.

- Up/down spinning reserve
  Based on the following constraint formula, even when a forecast error occurs between $\pm 2\sigma^{PV}$ in the current day operation plan, change of the start/stop state of the generator decided in the next day operation plan will not occur [28].

$$D - \alpha^{PV}\mu^{PV} + 2\sigma^{PV} \leq \sum_{i=1}^{N_G}(P_i^{max} \times X_i) \tag{7}$$

$$D - \mu^{PV} - 2\sigma^{PV} \geq \sum_{i=1}^{N_G}(P_i^{min} \times X_i) \tag{8}$$

- Ramp rate
  Ramp rate constraints are constraints that limit the rate of change of generator output.

$$|P_{it+1} - P_{it}| \leq \Delta P_i^{max} \tag{9}$$

- Transmission constraints
  Transmission line restrictions are restrictions to ensure that the transmission capacity $S_{lt}$ flowing through the transmission capacity limit $S_l^{max}$ does not exceed.

$$S_{lt} \leq S_l^{max} \tag{10}$$

- Minimum up/down time
  Minimum up and down time constraints are first formulated as mixed-integer linear expressions relying on binary variables associated with the startup, shutdown, and on/off states of generating units.

$$T_i^{on} \leq X_i^{on}(t) \tag{11}$$

$$T_i^{off} \leq X_i^{off}(t) \tag{12}$$

  The explanation of each variable is shown in abbreviation here.
- Demand response constraints

$$\sum_{t=1}^{24} D_{tb}^{net} = \sum_{t=1}^{24}(D_{tb}^{net} + DR_{tb}) \tag{13}$$

$$- DR_{tb}^{max} \leq DR_{tb} \leq DR_{tb}^{max} \tag{14}$$

where $DR_{tb}^{max}$ is the maximum amount of demand response. In this research, $DR_{tb}^{max}$ is 5% of the total load demand.

- Storage battery constraints

Hereinafter, the storage battery restriction will be described [29]. where ch is the charge energy of the storage battery, and dis is the discharge energy. Each storage battery output at time $t$ must be charge and discharge output within the range of $ch^{min}(dis^{min})$ and $ch^{max}(dis^{max})$. Also, charging and discharging cannot be performed beyond the limit of the state of charge (SoC).

$$SoC_{t,b} = SoC_{t-1,b} + ch_{tb} - dis_{tb} \quad \forall t \in T, \forall b \in N_B \tag{15}$$

$$SoC_{tb} \leq SoC_b^{max} \quad \forall t \in T, \forall b \in N_B \tag{16}$$

$$ch_{tb} \leq ch_b^{max} \quad \forall t \in T, \forall b \in N_B \tag{17}$$

$$dis_{tb} \leq ch_b^{max} \quad \forall t \in T, \forall b \in N_B \tag{18}$$

$$ch_b^{max} = \frac{SoC_b^{max}}{R} \quad \forall b \in N_B \tag{19}$$

In this section, assume that NaS batteries are used for storage batteries and $R = 6$ (see Table 2).

Also, the values of constraints in Equations (6), (9), (11) and (12) are described in Table 3. The value of $S_l^{max}$ in Equation (10) is shown in Table 4.

## 3. Simulation Conditions

The simulation conditions are shown in Table 1. "w/o ESS" is a case where a storage battery is not placed. Re-predictive re-planning that iteratively solves the optimization problem of the finite interval periodically in all cases (see Figure 2).

**Table 1.** Simulation conditions.

| Case | ESS | $\alpha^{PV}$ | DR |
|---|---|---|---|
| w/o ESS (Case 0) | × | 1 | × |
| 1 | ○ | 1 | × |
| 2 | ○ | 2 | × |
| 3 | ○ | 2 | ○ |

**Figure 2.** Rolling horizon.

The optimization period is 24 h, the sample time is 3 h, and the unit commitment at the time of prediction is updated every 3 h in real time. Table 2 shows the storage battery parameters used in this paper. Table 2 is quoted from the details of the storage battery published by NGK Insulators, Ltd., Nagoya, Japan, a Japanese company [30]. The power system assumed in this paper is shown in Figure 3.

**Table 2.** NAS battery parameter.

| NAS Battery | |
| --- | --- |
| Output | 0.8 MW |
| Capacity | 4.8 MWh |
| Cost | 1.92 Billion yen |
| Lifespan | 15 year |

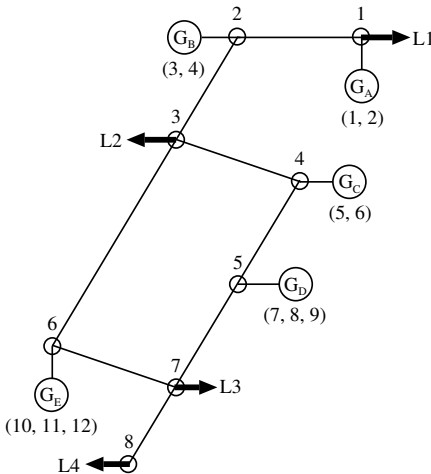

**Figure 3.** Power system model.

The generator parameters are shown in Table 3. Table 3 uses data published by "The Okinawa Electric Power", a Japanese electricity company in Okinawa Prefecture. In Table 3, *a*, *b* and *c* are the fuel cost characteristic constant of each generator. $G_A \sim G_E$ in Figure 3 are generator installation bus lines, and the numbers described in parentheses are the numbers of generator G in Table 3. Table 4 shows the transmission capacity assumed by the author. Resistance value R is simply 0 for DC power flow calculation [31]. Table 5 assumes the power demand of each load bus based on the population in Okinawa prefecture of Japan [32]. It is assumed that load demand can be predicted with high accuracy and only PV output is predicted. To verify the effectiveness of the proposed method in this research, we simulate a year from January 2017 to December 2017 (Analysis Stage). To solve this problem, MATLAB mixed integer programming (intlinprog) was used. Figure 4a,b show the actual demand of load demand and PV output ($\sigma^{PV} = 1$) for two years from January 2016 to December 2017, respectively. Figure 4a is the same source as [31]. Figure 4b is calculated using FFNN based on the data of solar radiation in 2015 in Okinawa prefecture in Japan [33].

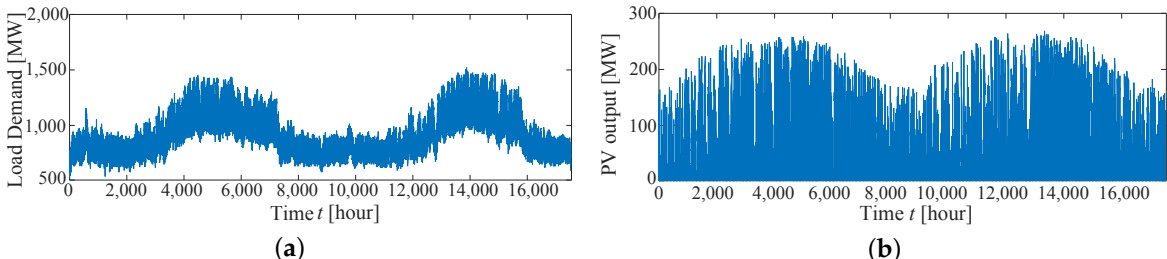

**Figure 4.** Load demand curve and PV output. (**a**) actual demand of load demand and (**b**) PV output.

Annual simulation is not efficient because it takes a lot of time. Therefore, by using clustering, we try to shorten the simulation time. Figure 5 shows the net load demand in 2016 (the demand from which PV output is subtracted from load demand).

**Table 3.** Generator's data.

|  | G1 | G2 | G3 | G4 | G5 | G6 |
|---|---|---|---|---|---|---|
| Use fuel | Coal | Coal | Oil | Oil | Coal | Coal |
| $P_{max}$[MW] | 220 | 220 | 125 | 103 | 156 | 156 |
| $P_{min}$[MW] | 84 | 84 | 60 | 50 | 60 | 60 |
| $a$ [yen] | 80,000 | 80,000 | 632,000 | 632,000 | 80,000 | 80,000 |
| $b$ [yen/MW] | 4000 | 4000 | 9200 | 9200 | 4000 | 4000 |
| $c$ [yen/MW$^2$] | 0.4 | 0.4 | 2.1 | 2.1 | 0.4 | 0.4 |
| $\Delta P_i^{max}$[MW] | 44 | 44 | 62.5 | 51.5 | 31.2 | 31.2 |
| $SUC$[yen] | 1,100,000 | 1,100,000 | 375,000 | 309,000 | 780,000 | 780,000 |
| $T^{on/off}$[h] | 8 | 8 | 6 | 6 | 6 | 6 |
|  | **G7** | **G8** | **G9** | **G10** | **G11** | **G12** |
| Use fuel | LNG | LNG | LNG | Oil | Oil | Oil |
| $P_{max}$[MW] | 251 | 251 | 35 | 125 | 60 | 103 |
| $P_{min}$[MW] | 122 | 122 | 17 | 60 | 30 | 50 |
| $a$ [yen] | 132,000 | 132,000 | 132,000 | 632,000 | 632,000 | 632,000 |
| $b$ [yen/MW] | 4400 | 4400 | 4400 | 9200 | 9200 | 9200 |
| $c$ [yen/MW$^2$] | 5.0 | 5.0 | 5.0 | 2.1 | 2.1 | 2.1 |
| $\Delta P_i^{max}$[MW] | 84 | 84 | 35 | 62.5 | 30 | 51.5 |
| $SUC$[yen] | 753,000 | 753,000 | 105,000 | 375,000 | 180,000 | 309,000 |
| $T^{on/off}$[h] | 8 | 8 | 4 | 6 | 4 | 6 |

**Table 4.** Transmission line parameters.

| From Bus | To Bus | R [pu] | X [pu] | Limits [MVA] |
|---|---|---|---|---|
| 1 | 2 | 0.0 | 0.1 | 300 |
| 2 | 3 | 0.0 | 0.1 | 480 |
| 3 | 4 | 0.0 | 0.1 | 280 |
| 3 | 6 | 0.0 | 0.1 | 250 |
| 4 | 5 | 0.0 | 0.1 | 480 |
| 5 | 7 | 0.0 | 0.1 | 500 |
| 6 | 7 | 0.0 | 0.1 | 500 |
| 7 | 8 | 0.0 | 0.1 | 700 |

**Table 5.** Load demand data.

| Load No. | L1 | L2 | L3 | L4 |
|---|---|---|---|---|
| Bus No. | Bus 1 | Bus 3 | Bus 7 | Bus 8 |
| Ratio | 9% | 24% | 24% | 43% |

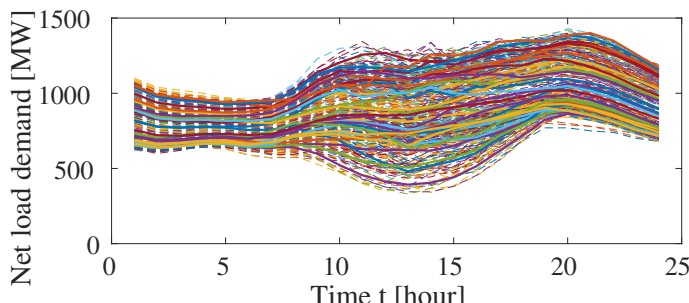

**Figure 5.** Net load demand.

The bold lines indicate demand classified into clusters. The number of clusters was set as 30 from the elbow method (Figure 6). The Sum of Squared Errors (SSE) is shown in the following equation.

$$SSE = \sum_{i=1}^{k} \{ \sum_{j=1}^{A_i} (y_i - \hat{y}_{i,j}))^2 \} \tag{20}$$

Here, $k$ is the number of clusters, $A_i$ is the total number of data classified as cluster $i$, $y_i$ is the cluster $i$, $\hat{y}_{i,j}$ In addition, the $j$ th data classified as cluster $i$.

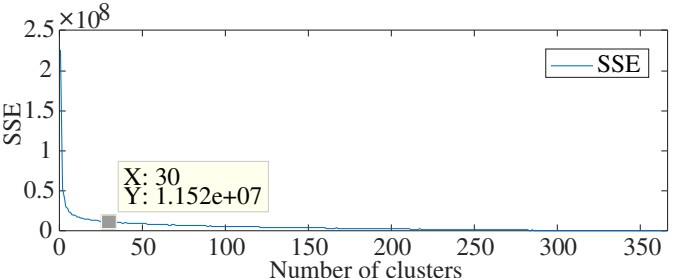

**Figure 6.** Elbow method.

## 4. Simulation Results

### 4.1. Evaluation Stage

Figure 7a–d show the probability density function of the maximum value ($SoC^{max}$) of the storage battery capacity in each cases. Figure 8 shows the average and standard deviation SOC of the maximum value ($SoC^{max}$) of the storage battery capacity of each bus line according to the annual simulation result. The standard deviation SOC in Figure 8 is taken as the optimum arrangement optimum capacity and is summarized in Table 6. In the case of bus 4 of Case 1 (see Figure 7a), the standard deviation $\sigma_4^{SoC}$ of $SoC_4^{max}$ is 64.77 MWh. From Table 2, since it is 4.8 MWh per unit, 64.77/4.8 = 13.4937 = 14 units. From Table 6, it can be read that the storage batteries are not arranged in the bus 1 and many are arranged in the bus 7 and 8. This is because of the load demand of the bus 1 in the power system of Figure 3 is small and there are enough thermal power generators. On the other hand, the load demand is large near the bus 7 and 8, and there are not many generators installed near them. At the peak of load demand, the capacity of the transmission line in the vicinity is congested, and it is necessary to start the low-efficiency generator around it, which increases the operation cost. Therefore, by arranging the storage batteries in the bus 7 and 8 and reducing the size of the load demand by the discharge power of the storage batteries, it is possible to reduce the cost.

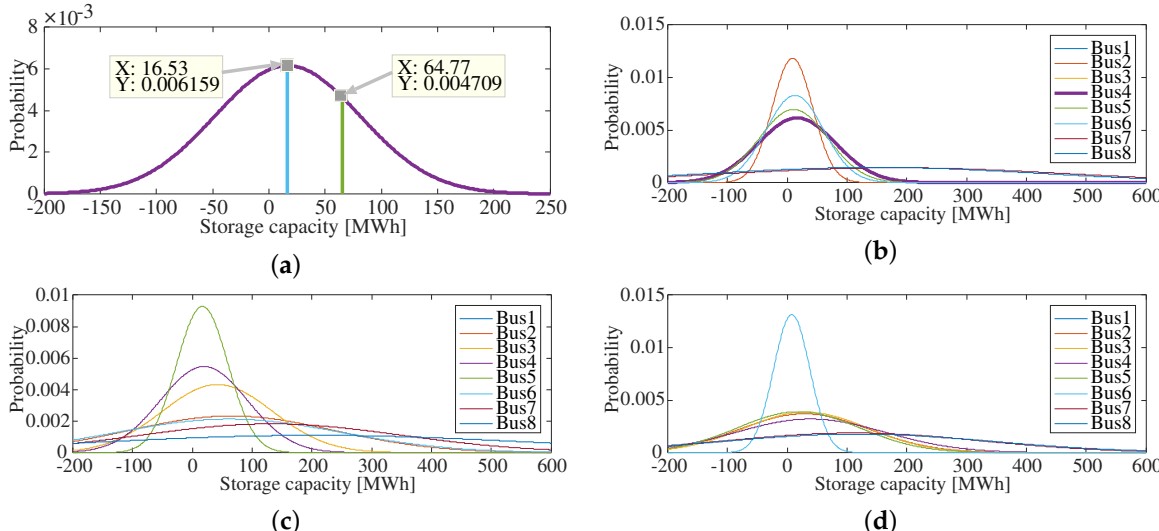

**Figure 7.** (**a**): PDF of Maximum battery capacity for 1 year in Case 1; (**b**): PDF of maximum battery capacity for 1 year in Case 1 at Bus 4; (**c**): PDF of Maximum battery capacity for 1 year in Case 2; (**d**): PDF of Maximum battery capacity for 1 year in Case 3.

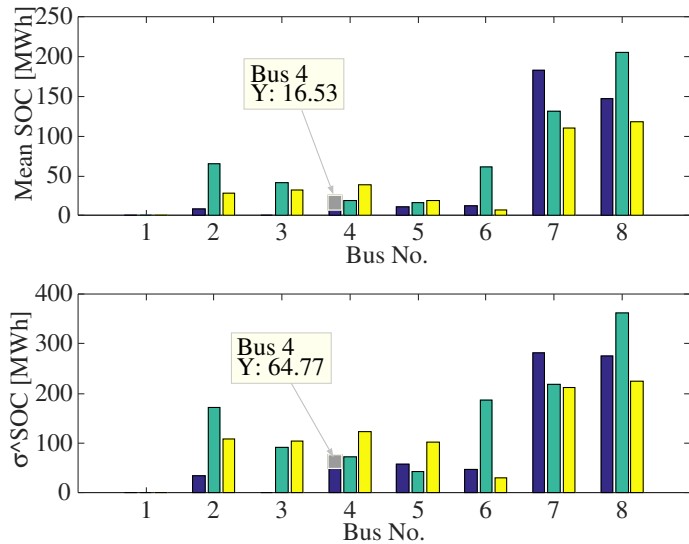

**Figure 8.** Mean and standard deviation of maximum battery capacity in each case (blue: Case 1, green: Case 2, yellow: Case 3).

**Table 6.** Optimal placement and optimum capacity.

| Bus No. | 1 | 2 | 3 | 4 | 5 | 6 | 7 | 8 | Total |
|---|---|---|---|---|---|---|---|---|---|
| Units (Case 1) | 0 | 7 | 0 | **14** | 12 | 10 | 59 | 58 | 160 |
| Units (Case 2) | 0 | 36 | 19 | 15 | 9 | 39 | 45 | 75 | 238 |
| Units (Case 3) | 0 | 22 | 22 | 26 | 21 | 6 | 44 | 47 | 188 |

*4.2. Analysis Stage*

Figure 9a shows the daily load rate. Figure 9b,c show the operating costs for each year in each case. Figure 10 shows the total cost in each case. Table 7 shows simulation results of Analysis Stage. From Figure 9a,b, it can be confirmed that the daily load rate is improved and the operation cost can be reduced through one year when the storage battery is arranged (Case 1). From Figure 9c, it can be confirmed that the introduction of PV throughout the year (Case 2, 3) can greatly reduce operating costs. Figure 10 and Table 7, when the storage battery is not considered (Case 0) and the case where the storage battery is arranged (Case 1), the total cost is reduced by $427.51 - 409.96 = 17.55$ billion yen I was able to achieve it.

**Table 7.** Simulation result.

| Case | Battery Cost [Billion Yen] | Operating Cost | Total Cost |
|---|---|---|---|
| 0 | 0 | 427.51 | 427.51 |
| 0 (w/o Day-212) | 0 | 413.32 | 413.32 |
| 1 | 20.35 | 389.62 | 409.96 |
| 2 | 30.58 | 357.12 | 387.69 |
| 3 | 24.15 | 357.83 | 381.99 |

In addition, it can be confirmed that reduction of $413.32 - 409.96 = 3.36$ million yen can be achieved even when compared with the case where Day - 212 where penalty cost occurred is excluded. Comparing the case where PV was introduced from $\sigma^{PV} = 1$ (Case 1) to $\sigma^{PV} = 2$ (Case 2), the storage battery cost increased from 2.05 billion yen to 3.058 million yen, but the total cost increased from 40.996 billion yen It can be confirmed that it can be reduced to 38.769 billion yen. When DR is not taken into account (Case 2) and DR is taken into account (Case 3), by considering DR, the storage battery cost can be reduced and the total cost can be reduced.

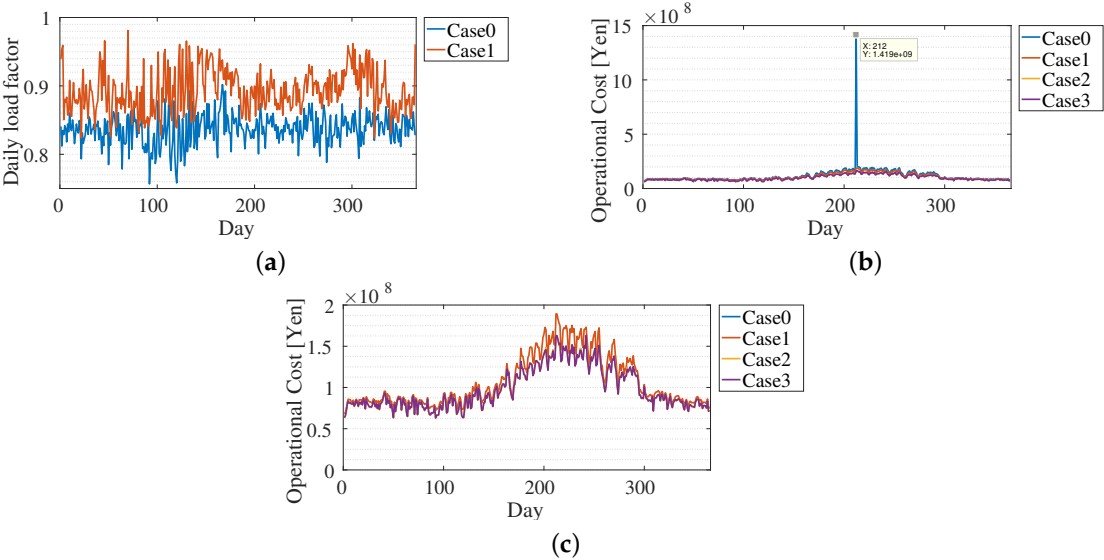

**Figure 9.** Simulation results. (**a**) Daily load factor; (**b**) Operation cost for 1 year and (**c**) Operation cost for 1 year (without Case 0).

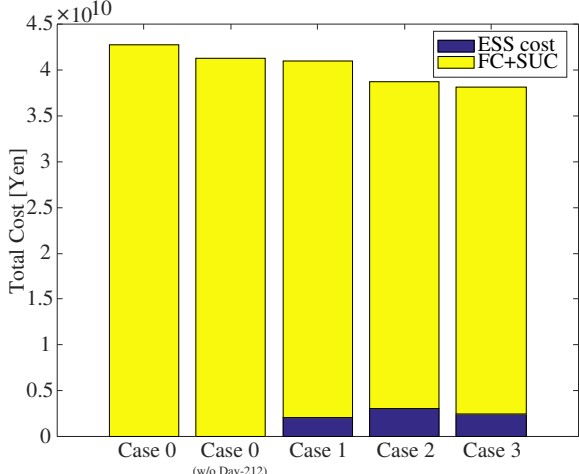

**Figure 10.** Total cost.

## 5. Conclusions

In this paper, we examined the optimal capacity and optimal placement method for storage battery installation in the power system where a large amount of photo-voltaic (PV) was introduced. Based on the probability density function, we calculated the capacity and location of the storage battery for economic improvement. Even when a large amount of renewable energy power generation equipment was introduced based on the calculated storage battery capacity and the operation plan introducing the placement, the operation was achieved without reducing the economic efficiency. At the same time, operation has been achieved which minimizes generator start-up costs, operating costs and storage battery costs. In addition, the introduction of demand response has also improved the operation cost. As future work, we will carry out system operation to meet various requirements by deciding the optimal capacity and placement of storage batteries considering the multi-objective functions.

**Author Contributions:** These authors contributed equally to this work.

**Conflicts of Interest:** The authors declare no conflict of interest.

## Abbreviations

The following notations are used in this manuscript:

Objective function and constraints:

| | |
|---|---|
| $b$ | Bus number |
| $ch_{tb}$ | Discharged energy of storage battery installed on bus $b$ |
| $ch_b^{max}$ | Battery rated output of the bus $b$ |
| $dis_{tb}$ | Discharged energy of storage battery installed on bus $b$ |
| $h$ | Expected life of the storage battery |
| $i$ | Thermal power Generator number |
| $n$ | Total number of days |
| $B$ | Total number of Bus |
| $C_{ens}$ | Cost of energy not served |
| $C_{SoC}$ | Battery cost per 1 MWh |
| $D$ | Total of load demand |
| $D_{tb}^{net}$ | Load demand at bus $b$ at time $t$ |
| $DR_{tb}$ | Amount of power transferred by demand response |
| $DR_{tb}^{max}$ | Demand response maximum amount |
| $ENS_t$ | Energy not served at hour $t$ |
| $FC_{it}$ | Fuel cost function |
| $IC_{SOC}$ | Storage battery cost per day |
| $N_B$ | Number of bus |
| $N_G$ | Number of the generators |
| $P_{it}$ | Output power of $i$thgenerator at time $t$ |
| $P_i^{min}$ | Minimum output limit of $i$th generator |
| $P_i^{max}$ | Maximum output limit of $i$th generator |
| $\Delta P_i^{max}$ | Maximum output change rate of generator $i$ |
| $\Delta PG_i^{max}$ | Ramp rate limits of generator $i$ |
| $R$ | Ratio of the storage battery capacity to the rated output |
| $SUC_{it}$ | Start-up cost function. |
| $SoC_{tb}$ | State of charge of storage battery installed on bus $b$ |
| $SoC_{b,i}^{max}$ | Maximum value of the storage battery capacity of the bus $b$ in the day $i$ |
| $S_{l,t}$ | Power flow on transmission line $l$ at time $t$ |
| $S_l^{max}$ | Maximum capacity of line $l$ |
| $T$ | Total time of day |
| $T_i^{on}$ | Minimum up time of $i$ |
| $T_i^{off}$ | Minimum down time of $i$ |
| $X_{it}$ | $i$th generator status at hour $t$ (1/0 for on/off) |
| $X_i^{on}(t)$ | Duration of continuously on of generator $i$ at time $t$ |
| $X_i^{off}(t)$ | Duration of continuously off of generator $i$ at time $t$ |
| $\alpha^{PV}$ | PV installation amount parameter |
| $\sigma^{PV}$ | Standard deviation of PV output prediction error |
| $\sigma^{SOC}$ | Standard deviation of foecast error |
| $\mu^{PV}$ | Output of PV |

Clustering variable:

| | |
|---|---|
| $k$ | Number of clusters |
| $y_i$ | Cluster $i$ |
| $\hat{y}_i$ | $j$ th data classified as cluster $i$ |
| $A_i$ | is the total number of data classified as cluster $i$ |

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
