# Peer review of "Optimum Capacity and Placement of Storage Batteries Considering Photovoltaics"

_sustainability, doi:10.3390/su11092556_

Round 1
Reviewer 1 Report
Abstract:
1. Please state the methodology and main results.
2. Line 2, change "photo-voltaic" to "photovoltaic"
Section 1
3. Divide the introduction into several paragraphs, to describe the background,
literature review, the contribution , and organization of the paper.
4. The aims of the work should be stated more clearly, and why the exisiting methods cannot solve the problem in hand.
Section 2
5. This section lists he objective function and constraints for optimization. However, more description should be added to explain each equation, the assumption used. Proper references should be added.
6. The meanings of each quantities should be explained in text, right after the equation, although some of them have been given in "Abbreviations".
7. For the battery constraints, what's the unit of "ch" and "dis"? power or current? This battery model sim
8. The values of the constraints are not given in the paper.
9. Have you considered the degradation of battery? That is, the performance of the battery in one year is different from the previous year.
Section 3
7. The source of the parameters in Table 1 to Table 4 should be given. Also, the source of the data in Figure 4, Figure 5 should be provided.
8. MILP and machine learning are mentioned in introduction as the methods used in the work. However, they are not discussed in main text. Hence, please explain briefly how do you implement the methods for optimization.
Abbreviations
9. The heading should be "Nomenclature" instead of "Abbreviations"
Generally, I can see the methods proposed by the authors can solve the optimization problem presented, and the results are reasonable. However, the contents are not well-presented. I suggest the author revise the paper by enriching the content, adding more details and description about the equations, methods, results and main findings.
Author Response
Abstract:
1. Please state the methodology and main results.
A. The methodology was derived from the three-stage battery capacity determination method in the literature [26]. In this article, we predict PV output in the first stage. In the second stage, the optimum capacity and number of storage batteries are obtained from the standard deviation. In the third stage, the simulation was performed for one year with the determined arrangement and capacity of storage batteries. As a result, it was possible to achieve the total cost minimization by the determined storage battery.
2. Line 2, change "photo-voltaic" to "photovoltaic"
A. I changed it. Section 1
3. Divide the introduction into several paragraphs, to describe the background, literature review, the contribution, and organization of the paper.
A. I changed the introduction configuration.
4. The aims of the work should be stated more clearly, and why the existing methods cannot solve the problem in hand.
A. The purpose of this work is to determine the optimal placement and capacity of storage batteries in order to minimize operating costs. In this work, comparison with the existing methodology [26] is not performed, and it will be necessary to examine the comparison of total cost and simulation time in the future.
Section 2
5. This section lists the objective function and constraints for optimization. However, more description should be added to explain each equation, the assumption used. Proper references should be added.
A. Added an explanation to the formulation.
6. The meanings of each quantity should be explained in the text, right after the equation, although some of them have been given in "Abbreviations".
A. In the part of storage battery restriction, I corrected it as the explanation was duplicated as you pointed out.
7. For the battery constraints, what's the unit of "ch" and "dis"? power or current? This battery model sim
A. "Ch" is the amount of charge [MW] in the storage battery. "Dis" is the discharged energy [MW] of the storage battery. I have stated in the storage battery constraints part of this article.
8. The values of the constraints are not given in the paper.
A. Also, the values of constraints in Equations (6), (9), (11) and (12) are described in Table3. The value of Slmax in equation (10) is shown in Table4.
9. Have you considered the degradation of battery? That is, the performance of the battery in one year is different from the previous year.
A. It does not consider the deterioration of the battery. The reason is that we are dealing with the problem simply. We think that it is necessary to consider if we think about a realistic implementation.
Section 3
7. The source of the parameters in Table 1 to Table 4 should be given. Also, the source of the data in Figure 4, Figure 5 should be provided.
A. Table 1 shows the conditions for each case I considered.
Table 2 is quoted from the details of the storage battery published by NGK Insulators, Ltd., a Japanese company.
This will be described as [30].
NGK Insulators, Ltd. Available online: https://www.ngk.co.jp/ ( accessed on 11 January 2017).
Table 3 uses data published by "The Okinawa Electric Power", a Japanese electricity company in Okinawa Prefecture.
This will be described as [31].
The Okinawa Electric Power Company, Incorporated (OEPC) Available online: https://www.okiden.co.jp/ (accessed on 11 January 2017).
Table 4 shows the transmission capacity assumed by the author. Resistance value R is simply 0 for DC power flow calculation.
Table 5 assumes the power demand of each load bus based on the population in Okinawa prefecture of Japan.
This will be described as [32].
Okinawa Prefectural Statistical Data Website Available online: https://www.pref.okinawa.jp/toukeika/index.html(accessed on 11 January 2017).
Figure 4a is the same source as [31]. Fig. 4b is calculated using feedforward neural network (FFNN) based on the data of solar radiation in 2015 in Okinawa Prefecture in Japan.
Figure 5 shows the 2016 load demand of Figure 4a minus the PV output of Figure 4b in parallel. MathWorks MATLAB Website Available online: https://jp.mathworks.com/help/deeplearning/ref/feedforwardnet.html (accessed on 11 January 2017).
The contents of these will be described in the text.
8. MILP and machine learning are mentioned in the introduction as the methods used in the work. However, they are not discussed in the main text. Hence, please explain briefly how do you implement the methods for optimization.
A. Details are described in section 2.
Abbreviations
9. The heading should be "Nomenclature" instead of "Abbreviations"
A. I tried to change it, but I understood that the heading is written as "abbreviation" because of the latex specification. It is necessary to install a new package in order to replace it with "Nomenclature".
If you must replace "Nomenclature", we will install the package and post it again.
The change is clearly described in the PDF.
Red text: added text
Blue text: deleted text
Green text: Changed text

Reviewer 2 Report
In this article, the authors present a research on examination of optimal capacity and optimal placement method for storage battery installation where photo-voltaic (PV) was introduced by using simulation. Overall, this manuscript has certain novelty. The reviewer believes it will attract attention from researchers in this area. However, the manuscript needs to be carefully revised and the reviewer has following questions that need the authors to clarify before this article can be considered to be published in Sustainability.
1. The English, grammar and format of this article need to be carefully polished before it can be considered to publish. For example, could the authors separate the introduction section with several paragraphs? It is odd to have one big paragraph as the introduction part.
2. Could the authors add the meaning of each equation in section 2? The reviewer found it is very confusing with just listing the equations.
3. Could authors add more discussions regarding the figures? For example, in figure 3, what are the Ga, Gb,…in the circle? What the arrows and symbols represent? It stays unclear to the readers. Could the author revise the caption of each figure and make it more specific.
4. For the simulation results section, the authors need to add more explanations and discussions regarding the results of each figure. For example, what are the differences between Figure b, c and d in Figure 7? What are the reasons for the differences?
Author Response
1. The English, grammar and format of this article need to be carefully polished before it can be considered to publish. For example, could the authors separate the introduction section with several paragraphs? It is odd to have one big paragraph as the introduction part.
A. Thank you for your advice. Introduction to the dissertation background, literature review, contributions, and composition.
2. Could the authors add the meaning of each equation in section 2? The reviewer found it is very confusing with just listing the equations.
A. Added the meaning of each equation in section 2.
3. Could authors add more discussions regarding the figures? For example, in figure 3, what are the Ga, Gb,…in the circle? What the arrows and symbols represent? It stays unclear to the readers. Could the author revise the caption of each figure and make it more specific.
A. GA ~ GE in FIG. 3 are generator installation bus lines, and the numbers in parentheses are the numbers of generator G in Table 3.
4. For the simulation results section, the authors need to add more explanations and discussions regarding the results of each figure. For example, what are the differences between Figure b, c and d in Figure 7? What are the reasons for the differences?
A. The caption in Figure 7 has been rewritten as follows.
(a)PDF of Maximum battery capacity for 1 year in Case 1. (b)PDF of maximum battery capacity for 1 year in Case 1 at Bus 4. (c) PDF of Maximum battery capacity for 1 year in Case 2. (d) PDF of Maximum battery capacity for 1 year in Case 3.
The change is clearly described in the PDF.
Red text: added text
Blue text: deleted text
Green text: Changed text

Reviewer 3 Report
The subject of the article is interesting and currently very much needed, but the article seems to be very casual.
Comments, questions and doubts:
In chapter 2.2 the restrictions are described in my opinion too briefly and laconically. Many symbols from the formulas are not explained and described in the text, although they are at the end of the abbreviation list.
Was the operating temperature taken into account in optimizing the location of storage batteries?
The article is about PV, and there is no PV in figure 3 and table 3. So where does PV appear in the system?
Figure 5 - What is marked with dashed lines? Do the colors of the lines have any meaning?
Figure 7 - How to understand the negative storage capacity on the axes?
The article requires, in my opinion, a more detailed description of the methodology used and simulation research, which will facilitate its more intuitive reception by the reader.
Author Response
In chapter 2.2 the restrictions are described in my opinion too briefly and laconically. Many symbols from the formulas are not explained and described in the text, although they are at the end of the abbreviation list.
A. Added a description of each equation in section 2.
Was the operating temperature taken into account in optimizing the location of storage batteries?
A. No, operating temperature is not taken into consideration. The reason is that we are dealing with the problem simply. We think that it is necessary to consider if we think about a realistic implementation.
The article is about PV, and there is no PV in figure 3 and table 3. So where does PV appear in the system?
A. It is assumed that each load bus is installed. The amount of power generation of PV was predicted beforehand by a neural network. It is assumed that PV is output at the same rate as the size of the load demand of each load bus as a whole, the amount of power generation at that time. That is, the output ratio of PV is the same as Table 4.
Figure 5 - What is marked with dashed lines? Do the colors of the lines have any meaning?
A. As this shows load demand data for 365 days in one figure, we used various colors and broken lines for clarity. There is no difference between each load demand.
Figure 7 - How to understand the negative storage capacity on the axes?
A. Fig. 7 shows the probability density distribution centered on the average of the storage battery SOC, and considering the deviation σ, such a negative value will be obtained. Therefore, negative values have no meaning.
The change is clearly described in the PDF.
Red text: added text
Blue text: deleted text
Green text: Changed text

Reviewer 4 Report
This paper presents the determination of storage battery placement and capacity considering one year is performed by three-step simulation based on probability density function. The organization and presentation of the paper is good. However, some improvement in the paper is necessary. My main points of concern are the following:
- Results do not bring substantive new insights. At least one novel direction is needed, either in theory, methodology or empirical. All three meet at medium level with no one direction excelling to novelty.
- Highlight your exact contribution which makes it different from the rest.
- The decision variable SoCmax in Eq. (1,2) is not stated. SoCmax would be a crucial computational burden.The coefficients have to be updated once the values of SoC changed and a power flow solution is needed for this update, resulting in time-consuming computational burden. If other approach is used, the author should elaborate this approach and analyze its computation efficiency.
- Fig. 1 is not clear.
- The paper do not mention "num of leaves (population)". How can we get this value?
- Explain the working principle of a ICsoc.
- The authors should give out more simulation results to verify their claimed merits.
- I suggest using the benchmark references in the various methods mentioned in this paper, i.e., the references in which such methods were proposed, e.g., "bat algorithm [A], particle swarm optimization [B], genetic algorithm, ANFIS etc.;
- It lacks references from others studies to prove why your work is novel. More recent references are required for the paper such as "Nguyen et al.,Determination of Optimal Location and Sizing of Solar Photovoltaic Distribution Generation Units in Radial Distribution Systems, Energies 2019" and " Kien et al., A Novel Social Spider Optimization Algorithm for Large-Scale Economic Load Dispatch Problem, energies 2019".
Author Response
- Results do not bring substantive new insights. At least one novel direction is needed, either in theory, methodology or empirical. All three meet at medium level with no one direction excelling to novelty.
- Highlight your exact contribution which makes it different from the rest.
A. For the above two problems.
I added the following sentences to the introduction of the text.
In this paper, we propose a new method of determining the optimal arrangement and capacity for storage batteries. By dividing the simulation into three stages, we determined the optimal placement and capacity of the storage battery. This is derived from the idea of the three-stage battery capacity determination method in the literature [26]. The first stage predicts PV output. In the second stage, taking into account the PV prediction error, the optimum capacity and the optimum number of storage batteries are obtained from the standard deviation. In the third stage, a one-year simulation is conducted to confirm the effectiveness of the determined battery placement and capacity.
- The decision variable SoCmax in Eq. (1,2) is not stated. SoCmax would be a crucial computational burden. The coefficients have to be updated once the values of SoC changed and a power flow solution is needed for this update, resulting in time-consuming computational burden.
If other approach is used, the author should elaborate this approach and analyze its computation efficiency.
A. We used clustering, selected representative days, and reduced the computational load. However, since the claims in this paper aim to minimize the total cost, we do not analyze the computational load. I also think that doing analysis in the future is useful.
- Fig. 1 is not clear.
A. Details are described in section 2.
- The paper do not mention "num of leaves (population)". How can we get this value?
A. Table 3 uses data published by "The Okinawa Electric Power", a Japanese electricity company in Okinawa Prefecture. This will be described as [31].
The Okinawa Electric Power Company, Incorporated (OEPC) Available online: https://www.okiden.co.jp/ (accessed on 11 January 2017).
- Explain the working principle of a ICsoc.
A. C_soc is the storage battery cost per 1 MWh and h is the expected life of the storage battery.
Storage battery cost is quoted from [30].
Table 2 is quoted from the details of the storage battery published by NGK Insulators, Ltd., a Japanese company.
This will be described as [30].
- The authors should give out more simulation results to verify their claimed merits.
A. As you pointed out, we plan to compare the simulation results with the conventional method [26] in the future.
- I suggest using the benchmark references in the various methods mentioned in this paper, i.e., the references in which such methods were proposed, e.g., "bat algorithm [A], particle swarm optimization [B], genetic algorithm, ANFIS etc.;
and
- It lacks references from others studies to prove why your work is novel. More recent references are required for the paper such as "Nguyen et al.,Determination of Optimal Location and Sizing of Solar Photovoltaic Distribution Generation Units in Radial Distribution Systems, Energies 2019" and " Kien et al., A Novel Social Spider Optimization Algorithm for Large-Scale Economic Load Dispatch Problem, energies 2019".
A. At present, we do not compare with conventional methods, and we think that it is necessary to make the effectiveness of the proposed method by doing in the future.
The change is clearly described in the PDF.
Red text: added text
Blue text: deleted text
Green text: Changed text

Round 2
Reviewer 1 Report
The quality of the paper has been improved. However, there are some further improvement should be done. The English presentation must be improved a lot, please ask a native English speaker to improve the English. Some comments are given below:
Page 1, line 21, remove "sdsa"
Page 2, line 46, define "DISCO"
Line 57, remove the heading "Contributions and Structure of This paper"
Once again, please re-examine all the equations and make sure each variable has been explained in the text. For example, NG, and NB in equation (3).
The summation symobl "Σ" in the equations is not correctly used in the paper. For example in (3), the content below Σ should be "i=1", "t=1", "b = 1", instead of "i","t","b"
More description and discussion must be given for the figures of simulation results. The discussion is too brief.
Author Response
Page 1, line 21, remove "sdsa"
A. It has been deleted.
Page 2, line 46, define "DISCO"
A. Specified DISCO.
Line 57, remove the heading "Contributions and Structure of This paper"
A. The headline has been deleted.
Once again, please re-examine all the equations and make sure each variable has been explained in the text. For example, NG, and NB in equation (3).
A. Variables not shown are shown in the appendix.
The summation symobl "Σ" in the equations is not correctly used in the paper. For example in (3), the content below Σ should be "i=1", "t=1", "b = 1", instead of "i","t","b"
A. I have corrected the specified part.
More description and discussion must be given for the figures of simulation results. The discussion is to brief.
A. I added the result of the evaluation stage.
I also made several other English corrections.
The change is clearly described in the PDF.
Red text: added text
Blue text: deleted text
Green text: Changed text

Reviewer 2 Report
The authors answered most of the concerns from the reviewer. The manuscript can be accepted in current form.
Author Response
We have made some corrections as pointed out by other reviewers.
The change is clearly described in the PDF.
Red text: added text
Blue text: deleted text
Green text: Changed text

Reviewer 4 Report
The paper was improved by authors.
Author Response

(The authors gave the same response as above.)

Round 3
Reviewer 1 Report
The previous questions have been successfully addressed.